# Age-Dependent Sex Differences in Perineuronal Nets in an *APP* Mouse Model of Alzheimer’s Disease Are Brain Region-Specific

**DOI:** 10.3390/ijms241914917

**Published:** 2023-10-05

**Authors:** Rayane Rahmani, Naiomi Rambarack, Jaijeet Singh, Andrew Constanti, Afia B. Ali

**Affiliations:** UCL School of Pharmacy, University College London, 29-39 Brunswick Square, London WC1N 1AX, UK; rayane.rahmani.19@ucl.ac.uk (R.R.); naiomi.rambarack.18@ucl.ac.uk (N.R.); jaijeet.singh.22@ucl.ac.uk (J.S.); a.constanti@ucl.ac.uk (A.C.)

**Keywords:** Aβ amyloid-β, AD Alzheimer’s disease, App β-amyloid precursor protein, CA1 Cornu Ammonis, LEC Lateral enthorinal cortex, PRS Presubiculum, PNNs Perineuronal nets, WGCNA Weighted gene co-expression network analysis

## Abstract

Alzheimer’s disease (AD) is the most common form of dementia, which disproportionately affects women. AD symptoms include progressive memory loss associated with amyloid-β (Aβ) plaques and dismantled synaptic mechanisms. Perineuronal nets (PNNs) are important components of the extracellular matrix with a critical role in synaptic stabilisation and have been shown to be influenced by microglia, which enter an activated state during AD. This study aimed to investigate whether sex differences affected the density of PNNs alongside the labelling of microglia and Aβ plaques density.We performed neurochemistry experiments using acute brain slices from both sexes of the *APP^NL-F/NL-F^* mouse model of AD, aged-matched (2–5 and 12–16 months) to wild-type mice, combined with a weighted gene co-expression network analysis (WGCNA). The lateral entorhinal cortex (LEC) and hippocampal CA1, which are vulnerable during early AD pathology, were investigated and compared to the presubiculum (PRS), a region unscathed by AD pathology. The highest density of PNNs was found in the LEC and PRS regions of aged *APP^NL-F/NL-F^* mice with a region-specific sex differences. Analysis of the CA1 region using multiplex-fluorescent images from aged *APP^NL-F/NL-F^* mice showed regions of dense Aβ plaques near clusters of CD68, indicative of activated microglia and PNNs. This was consistent with the results of WGCNA performed on normalised data on microglial cells isolated from age-matched, late-stage male and female wild-type and APP knock-in mice, which revealed one microglial module that showed differential expression associated with tissue, age, genotype, and sex, which showed enrichment for fc-receptor-mediated phagocytosis. Our data are consistent with the hypothesis that sex-related differences contribute to a disrupted interaction between PNNs and microglia in specific brain regions associated with AD pathogenesis.

## 1. Introduction

Alzheimer’s disease (AD) is the most common form of dementia, which affects 55 million people worldwide. This debilitating disease is associated with progressive memory loss related to amyloid-β (Aβ) plaques, neuroinflammation, microglial activation, and dismantled synaptic mechanisms that are time dependent, affecting specific brain regions differentially. 

To date, there is no disease-modifying cure for AD, as most available treatments offer only temporary symptomatic relief without attenuating the progressive neurodegeneration caused by the disease [1]. This is related to the lack of understanding of the etiopathogenesis associated with AD and the multi-factorial pathways contributing to AD pathogenesis that need further exploring.

A structure that has recently gained a large amount of interest in the AD research field due to its potential for novel AD pathologies is the perineuronal net (PNN). PNNs are important components of the brain’s extracellular matrix (ECM), with a critical role in synaptic stabilisation, contributing to the maintenance of memory [2,3,4]. PNNs, which are mainly associated with fast spiking, GABAergic, parvalbumin-expressing interneurons in the CNS, take shape by enmeshing around these neurons in the brain, creating a netlike structure composed of a hyaluronan (HA) backbone linked to chondroitin sulphate proteoglycans (CSPGs) [5]. Although controversial—as some studies report a decrease in PNN abundance in AD [6], while others have reported an increase [7] or no change at all [8]—the overall consensus is that there is emerging evidence to suggest that there is an altered role of PNNs that could be due to the impact or underlying cause of the neurodegeneration observed in diseases such as AD [9], which requires further investigation.

Previous studies have focussed on PNN-associated changes in PNN density in AD with the amyloid beta plaque pathology [10]. This pathology is characterised by the progressive accumulation of Aβ plaques in several brain regions, including the lateral entorhinal cortex (LEC) and the hippocampal CA1 region, responsible for the formation of episodic memory and storage [11,12]. These are the first regions to be affected severely during AD pathogenesis, despite the neighbouring presubiculum (PRS) region having been found to be an unscathed region in clinical AD [13,14,15,16].

A major player associated with the Aβ pathology is the microglia, which represent the principal immune cells in the CNS and have a role in the clearance of Aβ plaques. [17]. Furthermore, recent human genetic studies and data bases show microglia adopting key roles in the pathogenesis of AD that are linked to not only their immunological roles and activation, but to their interactions with the different elements of the CNS and synapses, which may directly impact memory and cognition [18,19].

Interestingly, recent evidence has also demonstrated a possible interconnection between PNNs and sexual dimorphism [20]. Epidemiology studies show that ~65% of dementia patients in the United Kingdom are women, which is much higher compared to the prevalence of the disease in males [21]. Some of the aspects that increase the risk and progression of AD and contribute to observed sex differences include different structural, physiological, and hormonal factors. Structurally, there is sexual dimorphism in brain size and volume transmission, whereas physiological factors, such as cardiovascular disease including hypertension, have been associated with neurodegeneration and cognitive-impairment-associated abnormal Aβ pathology [22]. Lastly, sex hormone imbalances in oestrogens, especially 17b-oestradiol (E2), have a role in regulating synaptic plasticity and promoting neural survival. Therefore, deficits in hormone levels, especially the sharp decline of E2 in post-menopausal women, are believed to increase the risk of AD in women [23]. It is imperative to investigate the relation between sex and PNNs, especially considering the lack of knowledge and evidence associated with how sex and PNNs are implicated in the progression, prevalence, and pathogenesis of AD.

Therefore, to address the lack of understanding of the etiopathogenesis of AD, the present study aimed to investigate the involvement of PNNs during the pathogenesis of AD during various stages of the disease, using neurochemistry experiments involving an APP knock-in mouse model of AD, age-matched to wild-type littermates at two different timepoints of the disease. We hypothesize that age-dependent sex-related differences exist in the interaction between PNNs and are associated with AD pathogenesis in specific brain regions. Furthermore, we hypothesize that dismantled microglia-PNN interactions will contribute to the AD pathogenesis at later stages of the disease. Since it has been established that activated microglia can alter the composition of the ECM, specifically through interactions with CSPGs—which, as mentioned, are components of PNNs [24]—we also employed bioinformatics computation tools to perform weighted gene co-expression network analysis (WGCNA) [25,26], to explore the relationship between the gene sets and clinical traits of microglia, on the one hand, and PNNs, on the other, and to gain a better understanding of their combined role in AD.

## 2. Results

The current study aimed to investigate whether there was an alteration of sub-brain region intensity of WFA, representing PNNs in early- (2–5 months) and late-stage (12–16 months) AD using an *APP* knock-in mouse model compared to wild-type litter mates, and whether there was a sex-related difference in the density of PNNs. The results below detail the analysis performed in the LEC, PRS, and the CA1 region of the hippocampus, which revealed interesting findings (Appendix A illustrates the plan of coronal brain sections and examples of whole-brain-section PNN labelling in these regions). However, we also imaged the density of PNNs in the hippocampal CA2 region of the hippocampus, which is thought to play a critical role in social recognition memory and has unique cellular and molecular properties that distinguish it from other areas [9]. The density of the PNN labelling in CA2 was consistent with the different ages and genotypes, suggesting no significant alteration during the pathogenesis of AD, and thus has been included as Appendix A. The magnitude of the PNN density is included in Table 1, where a direct comparison with the other brain regions can be observed.

### 2.1. Age-Dependent Density of PNNs Correlated with Sub-Brain Region

Our results show that the density of PNNs in wild-type and *APP^NL-F/NL-F^* mice is region-specific (Figure 1). The highest levels of PNNs were shown in the LEC and PRS regions compared to CA1 in both age groups studied, 2–5 months wild-type (*F*_2,24_ = 131.2; *p* < 0.0001) and APP mice (*F*_2,24_ = 120.8; *p* < 0.0001), as well as 12–16 months old wild-type (*F*_2,24_ = 83.90; *p* < 0.0001) and *APP^NL-F/NL-F^* mice (*F*_2,24_ = 79.94; *p* < 0.0001) (Figure 1).

The density of PNNs was observed to be significantly higher in the older cohorts studied for both sexes of *APP^NL-F/NL-F^* mice. For example, when comparing the abundance of PNNs between the two age windows used from *APP^NL-F/NL-F^* mice, there was an approximately 2-fold difference in the PNN density in CA1, as well as an approximately 2-fold difference in PNN density in the LEC in the older male cohort (*F*_1,42_ = 41.72; *p* < 0.0001), illustrating an age-dependent increase in the abundance of PNNs. This was consistent for the *APP^NL-F/NL-F^* females, where there was an approximately 2-fold higher density of PNN in CA1 and LEC regions in 12–16-month-old *APP* females when compared to 2–5-month-old *APP* female mice (*F*_1,42_ = 41.72; *p* < 0.0001). As for the wild-type mice, we noticed an insignificant higher level of PNNs related to age.

### 2.2. Sex Differences and PNN Alteration in Humanised Knock-In AD Mouse Model and Control Mice

To further investigate the sex differences associated with PNN density, we focused on the three brain regions CA1 (Figure 2A), LEC (Figure 2D), and PRS (Figure 2G). Two-way ANOVA test performed showed no sex differences in the density of PNNs between males and females for both 2–5-month-old wild-type mice (*F*_2,24_ = 1.138; *p* = 0.2967) and 2–5-month-old *APP^NL-F/NL-F^* mice (*F*_1,24_ = 1.937; *p* = 0.1768).

As for the role of sex in influencing the PNN density in the different brain regions of the older *APP^NL-F/NL-F^* mouse model of the AD cohort, our data suggest a sex-specific alteration in PNN density in LEC (*F*_1,16_ = 13.53; *p* = 0.0020) and PRS (*F*_1,16_ = 10.35; *p* = 0.0054). Results from performing a two-way ANOVA showed ~1.3-fold higher PNN density in *APP^NL-F/NL-F^* females when compared to their age-matched *APP^NL-F/NL-F^* male counterparts in both LEC and PRS (shown in Figure 2F,I).

Analysis of CA1 from immunoperoxidase labelling did not reveal a statistical sex difference in the abundance of the PNNs; therefore, to validate these results further, z-stack images from immunofluorescence labelling (that provided a better depth of tissue area analysed) were analysed. The data from z-stack confocal images revealed an approximately 2-fold higher PNN density in *APP^NL-F/NL-F^* males compared to age-matched *APP^NL-F/NL-F^* females (*F*_1,16_ = 8.3; *p* = 0.0106). Figure 3D shows a significant difference in PNN density when comparing older wild-type males and females to the age-matched *APP^NL-F/NL-F^* mice (*F*_1,16_ = 44; *p* < 0.0001), of which neuroanatomical details can be found in Table 1. Therefore, our results from immunoperoxidase and immunofluorescence analysis show sex-dependent alteration in PNN density during the late stages of AD that are region dependent.

### 2.3. The Density of PNNs Is Higher in the Late-Stage AD Mouse Model

For the older cohort representing the late stage of AD (12–16 months) in CA1 and LEC (Figure 2C and Figure 2F, respectively), our data show a higher PNN density in *APP^NL-F/NL-F^* mice when compared to age-matched wild-type mice. In the CA1 region, results from performing a two-way ANOVA test revealed an approximately 2-fold higher level in PNN density when comparing 12–16-month-old *APP^NL-F/NL-F^* male mice to the age-matched wild-type male mice (Figure 2C), and an approximately 2-fold higher level in PNN density when comparing 12–16-month-old *APP^NL-F/NL-F^* females to age-matched wild-type females in the same region (*F*_1,16_ = 45.55; *p* < 0.0001) (Figure 2C). This was consistent with the LEC region, where we found an approximately 2-fold higher level in PNN density for the males and a 2- fold higher abundance of PNNs for the females when comparing old *APP^NL-F/NL-F^* mice to age-matched wild-type mice (*F*_1,16_ = 96.24; *p* < 0.0001) (Figure 2F). No significant differences were found between wild-type and *APP^NL-F/NL-F^* mice in the PRS region in the 12–16-month-old mice (*F*_1,16_ = 2.253, *p* = 0.1528) (Figure 2I), suggesting that this region is not affected by AD.

### 2.4. Sex-Related Differences in the Interaction of PNNs and Activated Microglia with Dense Aβ Plaques

It has been suggested that sex differences play a role in the depletion of PNNs due to the activation of microglia in AD; therefore, we investigated whether PNN density was associated with regions of activated microglia and whether these components were in close proximity to dense Aβ plaques in the CA1 (Figure 3) of 12–16-month-old male and female *APP^NL-F/NL-F^* age-matched to wild-type mice. Microglia regulate PNN components, including CSPGs, which are bound and labelled by WFA, while CD68^+^ was used as a marker of activated microglia. As mentioned, in the CA1, we observed an approximately 2-fold higher density of WFA+ PNN in *APP^NL-F/NL-F^* males compared to *APP^NL-F/NL-F^* females (*F*_1,16_ = 8.3; *p* = 0.0106). We also found a significantly higher density of PNNs when comparing wild-type to age-matched *APP^NL-F/NL-F^* mice for both males and females, which was quantified as an approximately 4-fold higher density for males and an approximately 3-fold higher density for females (*F*_1,16_ = 44; *p* < 0.0001) (Figure 3D). This correlated with significantly higher density of Aβ plaques for the *APP^NL-F/NL-F^* mice compared to their age-matched wild-type mice, with a 14-fold increase in Aβ density for males and a 9-fold increase in density for the females *F*_1,16_ = 52.79, *p* < 0.0005) (Figure 3B). As for the CD68^+^ microglia quantification (Figure 3C), we found an approximately 2-fold higher CD68^+^ microglia density for the males and an approximately 3-fold higher density for the females when comparing 12–16-month-old wild-type mice to age-matched *APP^NL-F/NL-F^* mice (*F*_1,16_ = 85.05; *p* < 0.0001); however, no sex-related differences were observed when quantifying the Aβ plaques (*F*_1,16_ = 0.07330; *p* = 0.790), and the CD68^+^ microglia (*F*_1,16_ = 0.1660; *p* = 0.6891) in the CA1 region (Figure 3B,C). All comparison values are presented in Table 2.

To further investigate any association of sex differences in microglia in AD, we performed a WGCNA, which showed one microglial module (yellow) with significant correlations with the traits tissue, age, genotype, and sex (Figure 4A), indicating that the expression of the genes in this module is affected by changes in the named traits. The grey module represents unassigned genes and is disregarded. No genes listed in the perineuronal net term list GO:0072534 were present in this module. We performed gene enrichment analysis at FDR-corrected *p* < 0.05, which showed terms relating to infection and immune response as well as phagocytosis (Figure 4B).

## 3. Discussion

In this study, we investigated whether PNNs, components of the extracellular matrix with a critical role in supporting GABA expressing neurons, were altered during early- and late-stage AD and whether differences in the abundance of PNNs in AD-vulnerable regions were associated with differences in sex. Using the *APP^NL-F/NL-F^* mouse model of AD, which harbours human *APP* genes and mimics the age-dependent progression of AD, we found that the density of PNNs is brain-region-specific, with the highest PNN abundance in the LEC and PRS regions, which interestingly increases during natural ageing as well during the pathogenesis of AD. We also observed significant sex differences in the LEC, PRE, and CA1 regions. We focused on the CA1 region, as it indicated that older female *APP^NL-F/NL-F^* mice showed a lower level of PNNs compared to age-matched male *APP^NL-F/NL-F^* mice.

### 3.1. The Density of PNNs Is Sub-Brain-Region-Dependent

The density of PNNs varied according to the different brain regions. This is in accordance with previous findings that suggested differences in the distribution patterns of PNNs in the brain [27], as well as with more recent findings that showed that PNNs are not uniformly expressed in all brain regions [28]. These differences are believed to be associated with the diverse functionalities that are exhibited by PNNs in the brain, which encompass their neuroprotective role and their role in regulating synaptic plasticity, signalling, and memory, which are all implicated in distinct neurological diseases [4]. Our results demonstrated that the regions with the highest abundance of PNNs were the LEC and PRS in both the wild-type and *APP^NL-F/NL-F^* cohorts from both age windows. These findings correlate with the data from [27], where the PRS showed the highest frequency of labelled PNNs. Our results also showed no significant difference in the PRS regions of the 2–5-month-old and 12–16-month-old wild-type mice when compared to their respective age-matched *APP^NL-F/NL-F^* mice. However, we did observe a difference in the abundance pattern in older females in our AD model. The lack of change in the males corroborates the hypothesis that the PRS is devoid of plaques with dystrophic neurites in early-onset, late-onset sporadic, and familial AD [13,14,15,16], and in males of mouse models of AD [29], suggesting preservation of this region during the pathogenesis of AD. However, there is a lack of comparison made with data from tissue from females, and our data highlights this underrepresentation. Thus, our results reveal brain-region-specific differences in PNN density; however, to obtain a clearer understanding of these structures, further investigations will be required to determine the underlying cause of these differences in PNN density in these brain regions and how that can affect AD pathogenesis.

### 3.2. Late-Stage AD Significantly Alters the Density of PNNs

The present study suggests that the density of PNNs is age dependent, with the levels of PNN density increasing with age in both genotypes studied in several brain regions, which is consistent with previous studies performed in post-mortem brain tissue [30]. Interestingly, the LEC and CA1 regions are the first regions to be affected during early AD [31] and were the regions that showed significantly altered density of PNNs in later stages of AD, which corroborates the results of previous studies [32,33].

Overall, we observe a greater density of PNNs in our AD mouse model at the late stages of AD compared to the age-matched control mice. This change could be suggestive of the theory of a “compensatory protective mechanism” to prevent neurodegeneration and damage caused by the progression of AD [34] as a result of the accumulation of amyloid beta plaques [11]. Others have also suggested that PNNs have a neuroprotective role [1,5,35], which is further supported by studies that found elevated levels of hyaluronic acid, CPSGs, as well as tenascin proteins in AD models [35]. These are components of the ECM structure of PNNs [5,8,24]. Therefore, an increase in their levels could essentially be linked to an increase in the formation of PNNs.

### 3.3. Sex Differences and PNN Density

The present study demonstrates that sex is a factor affecting PNN density in different brain regions including the CA1, LEC, and PRS. This sexual dimorphism in PNN density may be connected to changes and differences in hormone levels between males and females and a possible connection between PNNs and oestrogens, which is understudied. This corroborates a previous study that showed that PNN-associated neurons in the mPOA (medial preoptic area) express oestrogen receptor α, which is suggested to influence the signalling mechanism affecting the assembly of the PNNs during the reproductive cycle in rodents [36].

Evidence also suggests a high density of oestrogen receptors in the hippocampus [37], and interestingly, they are colocalised extensively with parvalbumin (PV) cells in various brain regions including the hippocampus [38]. This suggests a potential connection of oestrogen with PNNs that encapsulate PV cells, and a mechanism whereby oestrogen can regulate neuronal excitability via inhibitory interneurons, which aligns with previous studies that hypothesise that oestrogen receptors influence hippocampal synaptic plasticity and neural signalling, which may affect cognition and memory [39].

Furthermore, oestrogens are thought to have several roles in AD [23], including neuroprotection and influencing the regulation of amyloid beta pathology by reducing the levels of Aβ and preventing its accumulation [40], which are functions comparable to those of PNNs [5]. Therefore, it may be plausible that males require a higher abundance of PNNs in brain regions such as the CA1 to obtain neuroprotection against factors such as oxidative stress, which they are more susceptible to (Griffiths et al., 2019) [20], as they do not have the potential extra protection that can be provided by the oestrogen hormones. The lack of experimental connection between PNNs and oestrogens is a limitation of the current study; it needs to be investigated further and could potentially prove a novel approach to understanding the sex-related differences in AD prevalence.

### 3.4. The Effect of Sex Differences in the Interaction of Microglia and PNNs

Microglia have a complex association with PNNs. They are involved in the regulation of PNN density in the normal brain through the remodelling of the ECM structural components, including CSPGs, tenascins, and hyaluronan and proteoglycan link proteins [41,42]. They also contribute to the upregulation of these components during neuronal injury [43,44]. However, microglia are not the primary facilitators of this role, as during their depletion, PNN density increases [44,45,46,47]. Thus, the microglial function in the turnover of PNNs appears to be more focused on degradation. Activated response microglia (ARM) have been implicated in the extensive loss of PNNs in AD as PNN components have been observed within microglia both in the mouse model and in humans [18].

Many recent studies have investigated the sex-related differences affecting the abundance and activation states of microglia, from development to maturation, under various neuroinflammatory conditions [48,49,50]. Microglial cells isolated from female mouse models of AD shift from a homeostatic state towards an activated response (ARM) state faster than age-matched male mice [51]. This change involves more glycolytic activity and impaired phagocytosis in female mice, which causes reduced amyloid clearance in comparison with males. The sex-related changes in microglial morphology and increased Aβ plaque density were consistent in post-mortem cortical tissue from individuals with AD and age-matched controls [52]. Our results in mice are consistent with these findings and show higher levels of dense Aβ plaques in aged female mice. We observed no sex-related differences in CD68+ microglia; however, morphological changes were not investigated. Due to sex differences affecting microglial function in disease, we performed WGCNA on a publicly available dataset to assess associations with microglial genes and sex as a trait.

Our WGCNA also revealed a module of genes with a weak correlation with sex, meaning that their expression is associated with differences between sexes. Annotation of this set of genes using the KEGG pathway showed an enrichment for Fc-gamma receptor-mediated phagocytosis. Fc-gamma receptors (FcγRs) are receptors on the surface of cells that elicit immune responses including phagocytosis. The expression of FcγRs on microglial cells is reported to be increased during ageing and AD [53,54,55], but how these receptors are associated with microglial clearance of Aβ plaques and PNNs warrants further investigation.

## 4. Materials and Methods

### 4.1. Mouse Animal Procedures

All the procedures in this study were carried out in accordance with the British Home Office regulations under the Animal Scientific Procedure Act 1986, under the project licence (awarded in March 2018) PPL: P1ADA633A held by the principal investigator, Prof. Afia Ali. All procedures were approved by both internal and external UCL ethics committees, and in accordance with the ARRIVE guidelines for reporting experiments involving animals [56]. A total of 50 animals (disease model and wild-type) were used in this study. The animals had ad libitum access to food and water and were reared in cages with a maximum of five inhabitants, with a day:night cycle of 12 h:12 h.

The knock-in *APP^NL-F/NL-F^* AD mouse model was used for this study [57]. This mouse model was chosen because it faithfully reproduces the effect of AD Aβ pathology without overexpression artefacts in a time-dependent manner. The *APP^NL-F/NL-F^* model has two familial AD (FAD) mutations: KM670/671NL (Swedish) and I716F (Iberian). The former increases β-site cleavage of APP to produce elevated amounts of both Aβ40 and Aβ42, whereas the latter promotes γ-site cleavage at C-terminal position 42, thereby increasing the Aβ42/Aβ40 ratio in favour of the more hydrophobic Aβ42, as seen in clinical AD. Thus, the *APP^NL-F/NL-F^* mouse model shows Aβ accumulation and related pathology in an age-dependent manner, with initial accumulation shown at 6 months [57]. The knock-in line was crossed with C57BL/6 mice, and the resulting heterozygous pairs were used for breeding but excluded from experiments. Only male and female *APP^NL-F/NL-F^* and age-matched wild-type (C57BL/6) mice, aged between 12 to 16 months, were included.

Animals were genotyped via standard polymerase chain reaction using the following four primers: 5′-ATCTCGGAAGTGAAGATG-3′, 5′-TGTAGATGAGAACTTAAC-3′, 5′-ATCTCGGAAGTGAATCTA-3′, and 5′-CGTATAATGTATGCTATACGAAG-3′, as previously described [57].

### 4.2. Tissue Collection and Preparation

#### Mouse Brain Tissue

Tissue preparation was carried out as previously described [31,58]. All experiments were performed single-blinded. Mice were deeply anaesthetized using inhalation of isoflurane 4% followed by intraperitoneal injection of 60 mg/kg phenobarbitone. The level of anaesthesia was monitored using pedal and tail pinch reflexes; rate, depth, and pattern of respiration through observation; and the colour of mucous membranes and skin. The mice were then perfused transcardially with artificial cerebrospinal fluid (ACSF) containing: (in mM) 248 sucrose, 3.3 KCl, 1.4 NaH_2_PO_4_, 2.5 CaCl_2_, 1.2 MgCl_2_, 25.5 NaHCO_3_, and 15 glucose, bubbled with 95% O_2_ and 5% CO_2_.

The brains were immediately fixed after perfusion in 4% paraformaldehyde and 0.1% glutaraldehyde in 0.1M phosphate buffer for 24 h prior to sectioning.

### 4.3. Neuroanatomical Procedures and Analysis

#### Immunohistochemical Procedures and Analysis

Hippocampal coronal slices were sectioned at 70 µm thickness using a vibratome (Leica, Munich, Germany) from the same region of CA1 in reference to mouse brain atlases (Allen mouse brain atlas). Appendix A illustrates example representative whole brain sections that have been sectioned and imaged for analysis for LEC, PRS, CA1, and CA2.

The brain sections were incubated in 0.1M phosphate buffer solution (PBS) for 24 h on a microplate shaker (VWR, UK). Sections were permeated using 0.3% tris-buffered saline and Triton (TBS-T) solution and stained according to the Amylo-Glo protocol [59]. Sections were subsequently washed using TBS-T solution before the blocking procedure using 20% animal serum in PBS. Incubation in primary antibodies was performed for 48 h at 4 °C, and subsequent incubation in secondary antibodies was for 3 h at room temperature (Table 3). Following incubation in secondary antibodies, the sections were mounted with Vectashield (Vector Laboratories, Bridgend, UK). For immunoperoxidase analysis, the slices were incubated in avidin-biotin-horseradish peroxidase complex (Vector Laboratories, Bridgend, UK) solution, processed with DAB, and subsequently dehydrated and mounted (Khan et al., 2018). Sections treated for immunoperoxidase were imaged at ×10 and ×20 objectives using a light microscope (Leica, Munich, Germany). The sub-brain regions of interest that were imaged include the CA1, CA2, LEC, and PRS. WFA quantification from immunoperoxidase-stained images aimed to quantify the number of PNNs by measuring the density/mm^3^ using ImageJ software (1.5.3 version). DAB-stained pictures were taken under ×200 magnification using a light microscope. Pictures were processed by colour deconvolution and “H-DAB” to prevent staining artifacts and improve quantification. ROIs were carefully selected and were located using the manual joystick through the ×20 objective lens by systematically searching the slice and consistently evaluating the location with reference to an appropriate mouse atlas for all the brain regions studied including CA1 (stratum oriens, pyramidal layer, and stratum radiatum), LEC (layer 2 and 3), and PRS (layers 1 and 2, also illustrated in Figure 1, indicated by red dashed line). The numbers of PNNs within each ROI were counted using the cell counter function in ImageJ, and the total number obtained per region was divided by the volume of the section to determine cell density in cells/mm^3^.

### 4.4. Confocal Microscopy

To provide further assessment and in-depth quantification of PNNs in the CA1 region of the hippocampal regions, confocal microscopy analysing Z-stack immunofluorescence images was performed. From each brain section, Z-stacks at ×20 objective were taken using the Zeiss LSM880 confocal microscope (Zeiss, Baden-Württemberg, Germany) in unison with the Zeiss Zen Black imaging software from the CA1. The regions of interest (ROI), including CA1 (including stratum oriens, stratum pyramidale, and stratum lacunosum), were located using the manual joystick through the ×20 objective lens by systematically searching the slice and consistently evaluating the location with reference to an appropriate mouse atlas. Z stack images were taken at a resolution of 1024 × 1024 pixels with 12–14 Z steps through the depth of the slice and with application of appropriate filters to complement secondary antibody fluorescence: DAPI (405 nm), Alexa 488 (488 nm), Alexa 555 (555/565 nm).

Image analysis was performed with ImageJ software (1.5.3 version) using an automated macro specific to each quantification. The Z-stack images were processed at maximum intensity projection and split into the constituent colour channels. Following this, the IsoData auto thresholding method was used to demarcate signal from background and produce the ROI (Huang and Wang, 1994). The Analyse Particles function was then used (PNN size = size = 100.00–infinity, dense Aβ plaques = size = 50.00–infinity, and CD68+ microglia = size = 0.00–infinity). PNN and CD68+ structures were calculated using % area, which divides the area of all the particles that meet the minimum size limit by the total area of the image field [60,61]. To compare dense plaque accumulation, average area was used as the measure for Aβ structures, which gave the average area of the dense plaques within the field.

### 4.5. Co-Expression Expression Analysis

Weighted gene co-expression network analysis [62] was performed on cortical and hippocampal microglia isolated from 12- and 21-month-old age-matched male and female *APP^NL-G-F^* and C57Bl/6 wild-type mice accessed from a publicly available dataset GEO: GSE127892 [51]. Sala Frigerio et al. [51] analysed over 10,000 microglial cells at four different timepoints; however, in this study we focused on the two later stages. The raw data were processed and normalised using the variance-stabilising transformation method from the *DESeq2* package [63]. The WGCNA *blockwiseModules* function was performed on the normalised counts at a soft-threshold power of β = 3 with a minimum module size of 50 to identify networks correlated with traits of interest: tissue, age, genotype, and sex. ShinyGO 0.77 was then used to perform gene enrichment analysis on the modules of interest using the KEGG (Kyoto Encyclopedia of Genes and Genomes) pathway database [64,65,66].

### 4.6. Statistics

The statistical analysis was performed using GraphPad Prism version 9.0 for Windows and Microsoft Excel. Based on the differences observed between control and diseased data sets obtained in our preliminary studies, an *n* ≥ 5 was ideal for this study in order to reveal a statistical difference of >80% power, assuming a 5% significance level and a two-sided test for neuroanatomy experiments.

Various statistical tests were performed depending on the parameters used, and each figure legend details the specific statistical test used. Before performing any statistical test, the normality of the raw data was verified using the Shapiro–Wilk test and a ROUT test to identify potential outliers. In almost all of the cases, no outliers were identified. Where an outlier existed, it was removed from the data pool via the software. When comparing between three or more data sets from two genotypes, a one-way analysis of variance (ANOVA) was performed. When a second factor was taken into account, for example age, a two-way ANOVA was used. After any ANOVA, a post hoc test for multiple comparisons was applied. All *p*-values below 0.05 were considered significant, and asterisks were added to the presentation of the data as follows: * *p* < 0.05, ** *p* < 0.01, *** *p* < 0.001, **** *p* < 0.0001.

To assess the significance of the differences between groups, F-values were also taken into consideration. There are displayed as (F values_(dfn,dfd)_; *p*-value), where dfn stands for degrees of freedom numerator, and dfd stands for degrees of freedom denominator. As for the interpretation, larger F values were considered significant, whereas smaller F values were considered non-significant.

All figures displaying error bars represent ± the standard error of the mean. The “*n*” is given as the number of observations and is equal to the number of animals used, unless otherwise stated.

## 5. Conclusions

In summary, we report in this study that the density of PNNs is altered in the LEC, PRS, and hippocampal region CA1 during the late stages of AD. We reported a higher level of PNNs in late-stage *APP^NL-F/NL-F^* mice compared to control mice. In addition, we also show that sex and age are both factors that influence the density of PNNs in early- and late-stage AD.

## Figures and Tables

**Figure 1 ijms-24-14917-f001:**
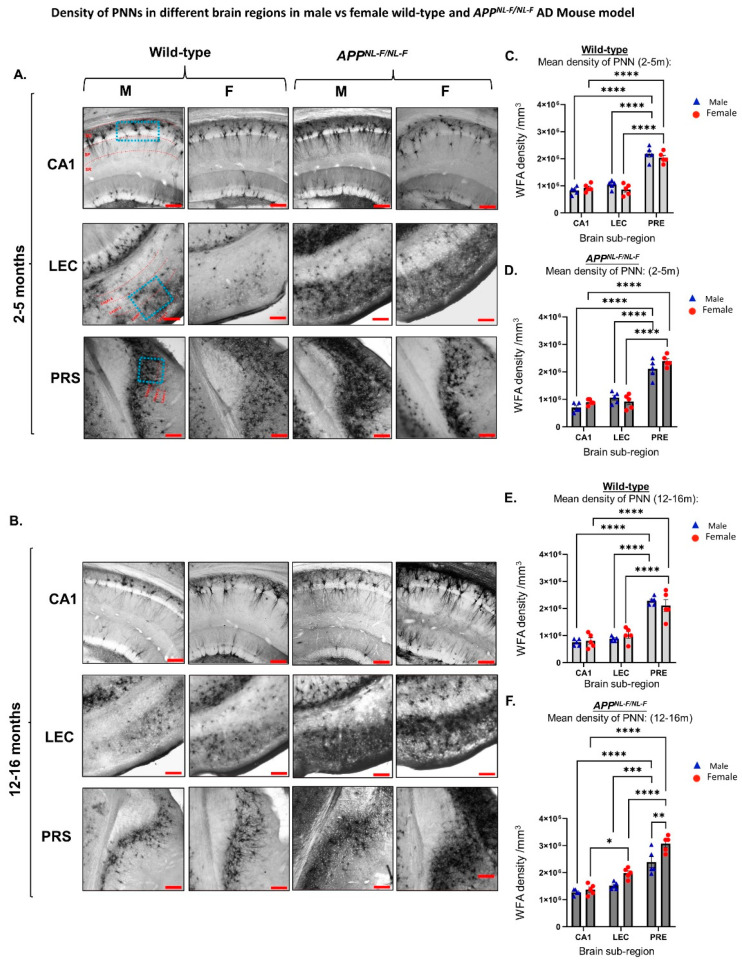
Density of PNNs is region-, age-, and sex-dependent. (**A**,**B**) Representative brightfield images taken at x100 magnification of the hippocampal CA1 region, LEC, and PRS from 2–5-month- and 12–16-month-old wild-type mice age-matched to the *APP^NL-F/NL-F^* mouse model. Scale bar 20 µm. (**C**,**D**) and (**E**,**F**) Graphs show quantitative data for the density of Wisteria floribunda agglutinin (WFA), indicative of the density of PNNs in CA1, LEC, and PRS from wild-type mice age-matched to the *APP^NL-F/NL-F^* mice at 2–5 months 12–16 months, respectively. (Two-way ANOVA corrected for multiple comparisons with post hoc Tukey’s test, * *p* < 0.05, ** *p* < 0.01, *** *p* < 0.001, **** *p* < 0.0001, *n* = 5 animals per cohort.).

**Figure 2 ijms-24-14917-f002:**
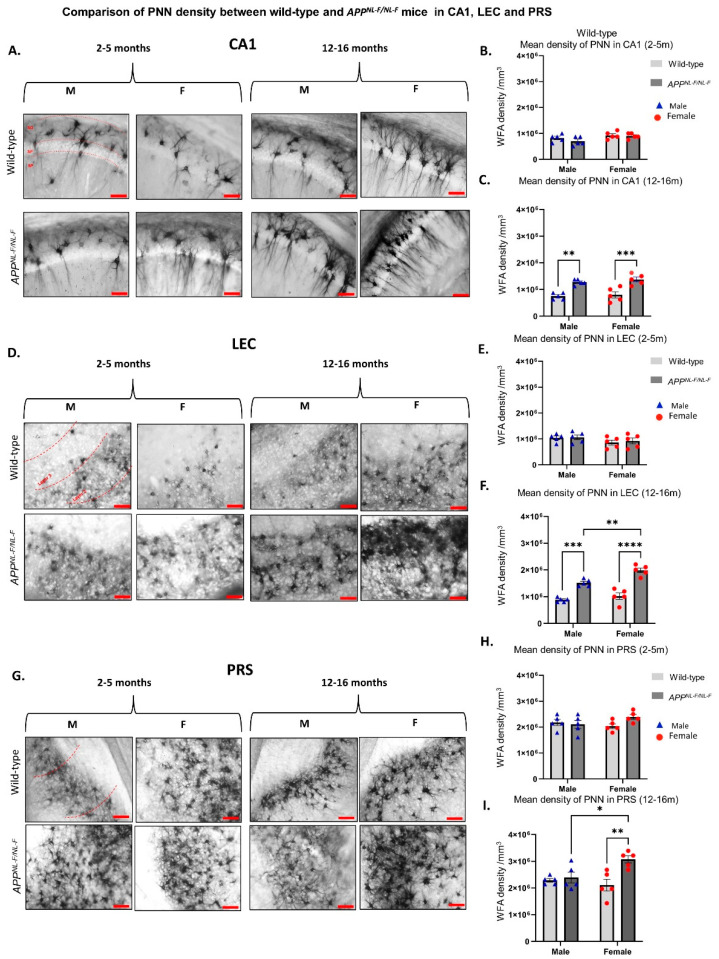
Change in the density of PNNs in the *APP^NL-F/NL-F^* mouse model of AD in different cortical regions. (**A**,**D**,**G**) Representative brightfield images taken at ×200 magnification from 2–5-month- and 12–16-month-old wild-type mice, age-matched to *APP^NL-F/NL-F^* mice from CA1, LEC, and PRS, respectively. Scale bar 10 µm. (**B**,**C**), (**E**,**F**), and (**H**,**I**) Graphs show the density of WFA, indicative of PNN density in CA1, LEC, and PRS in both genotypes at both age windows, 2–5 months and 12–16 months, respectively. Stratum oriens (SO), Stratum pyramidale (SP), stratum radiatum (SR) (Two-way ANOVA corrected for multiple comparisons with post hoc Tukey’s test, * *p* < 0.05, ** *p* < 0.01, *** *p* < 0.001, **** *p* < 0.0001, *n* = 5 animals per cohort.).

**Figure 3 ijms-24-14917-f003:**
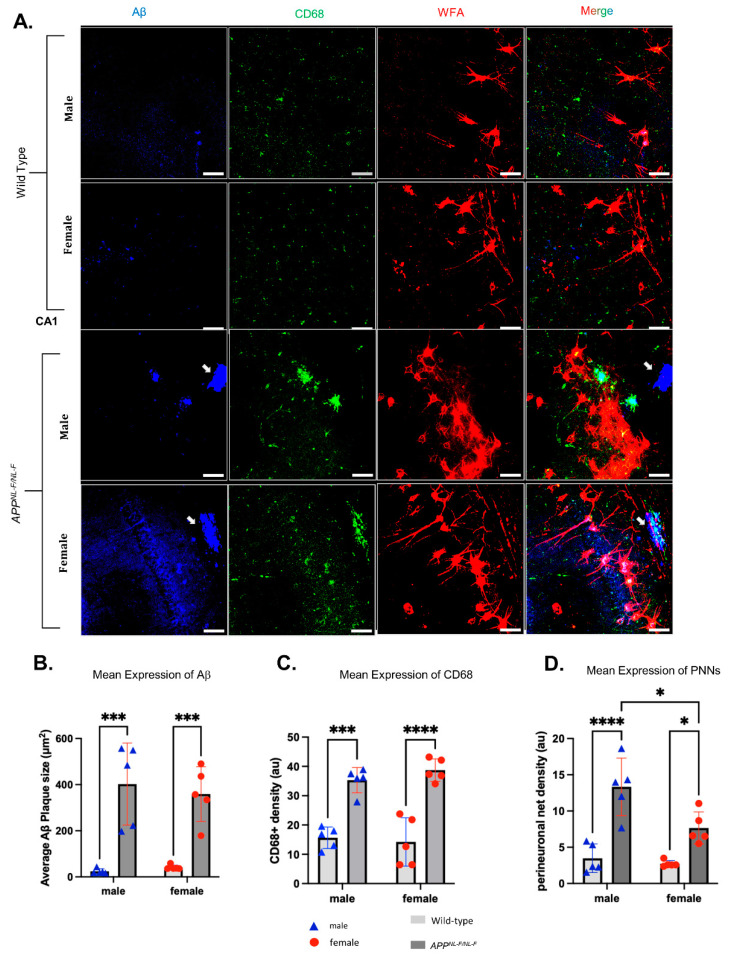
Density of PNNs and dense Aβ plaques are influenced by sex differences. (**A**) Representative images derived from confocal imaging of co-stained PNNs with WFA (red, Alexa Fluor 555, streptavidin), amyloid-β plaques Amylo-Glo (blue), and CD68+ microglia (green, Alexa Fluor 488) in the CA1 region using coronal brain sections of 12–16-month-old male and female wild-type and *APP^NL-F/NL-F^* mice at ×200 magnification. Scale bar 50 µm. (**B**–**D**). The white arrows in Figure (**A**) point to amyloid beta plaques in the APP mouse model. The results showed greater density of PNNs in male *APP^NL-F/NL-F^* mice compared with age-matched female mice in the CA1. (Two-way ANOVA corrected for multiple comparisons with post hoc Tukey’s test. * *p* < 0.05, *** *p* < 0.001, **** *p* < 0.0001, *n* = 5 animals per cohort).

**Figure 4 ijms-24-14917-f004:**
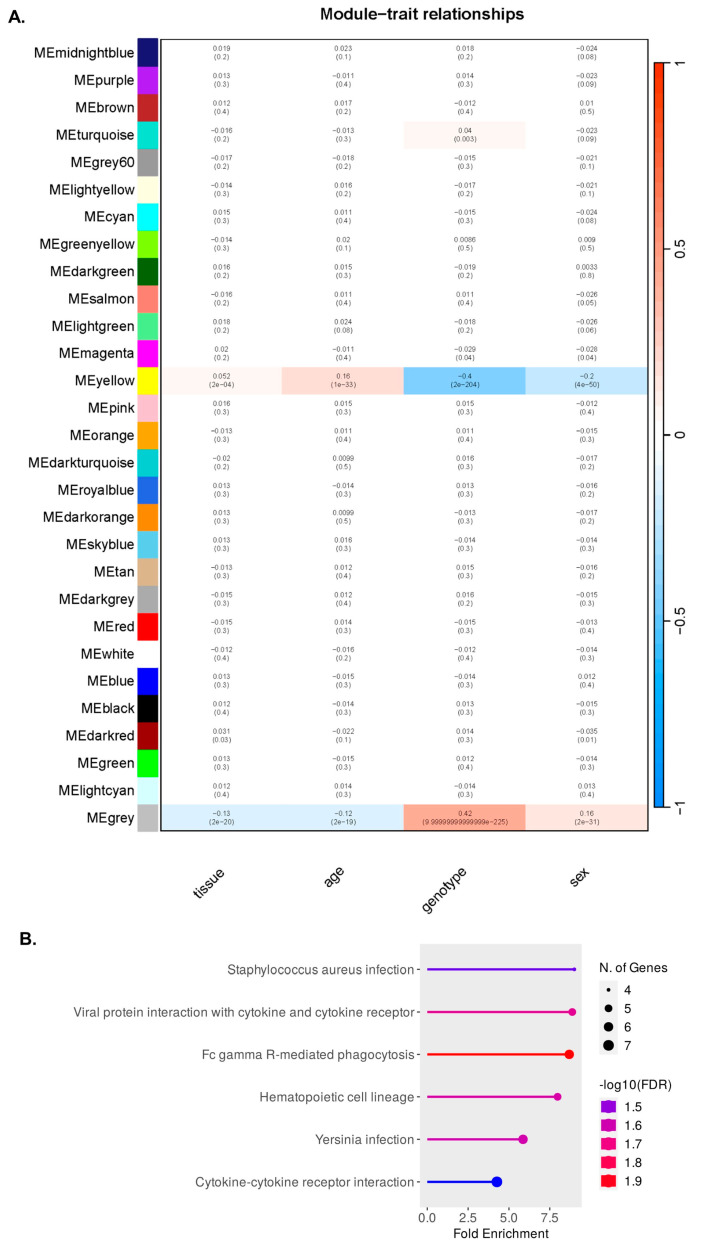
Sex differences associated with differential expression of genes relating to PNNs (**A**) Heatmap derived from WGCNA on microglial cells isolated from both the cortex and hippocampus of age-matched, late-stage (12–16 months) wild-type and App^L-G-F^ male and female mice (Sala Frigerio et al., 2019) showing one module (yellow) with weak correlations across traits: a positive correlation with tissue (cor= 0.052, *p*-value 2 × 10^-4^ ), a positive correlation with age (cor = 0.16, *p*-value = 1 × 10^-33^ ), a negative correlation with genotype (cor = − 0.4, *p*-value = 2 × 10^-4^ ), and a negative correlation with sex (*p*-value = cor = −0.2, *p*-value = 4 × 10^-50^ ). The correlation values of all modules with the traits are indicated by the red-to-blue scale of 1 to −1, respectively, with the *p*-values for the confidence of association with these traits also presented below. The grey module represents unassigned genes. (**B**) A ShinyGO-derived gene ontology analysis using the KEGG pathway on the yellow module of interest showing an enrichment for Fc-gamma receptor-mediated phagocytosis among other terms.

**Table 1 ijms-24-14917-t001:** Density of WFA, representing PNN density measured from immunoperoxidase staining and images of brain subregions CA1, LEC, and PRS obtained at ×20 magnification using the light microscope. Values represent the cell density in cells/mm^3^ and are stated as mean ± SEM. * denotes significant difference (*p* < 0.05, *n* = 5 animals per cohort) between wild-type control and *APP^NL-F/NL-F^* mice.

Region	Age	SEX	Wild-Type	*APP^NL-F/NL-F^*
			Mean ± SEM (Density of PNNs/mm^3^)	Mean ± SEM (Density of PNNs/mm^3^)
CA1	(2–5 months)	Male	0.83 × 10^6^ ± 6.37 × 10^4^	0.70 × 10^6^ ± 7.50 × 10^4^
		Female	0.93 × 10^6^ ± 6.37 × 10^4^	0.90 × 10^6^ ± 4.68 × 10^4^
	(12–16 months)	Male	0.75 × 10^6^ ± 5.59 × 10^4^	1.28 × 10^6^ ± 4.68 * × 10^4^
		Female	0.80 × 10^6^ ± 11.6 × 10^4^	1.38 × 10^6^ ± 8.84 * × 10^4^
LEC	(2–5 months)	Male	1.04 × 10^6^ ± 6.78 × 10^4^	1.06 × 10^6^ ± 9.27 × 10^4^
		Female	0.86 × 10^6^ ± 9.27 × 10^4^	0.92 × 10^6^ ± 11.6 × 10^4^
	(12–16 months)	Male	0.88 × 10^6^ ± 3.74 × 10^4^	1.52 × 10^6^ ± 5.83 * × 10^4^
		Female	1.02 × 10^6^ ± 12.0 × 10^4^	1.98 × 10^6^ ± 8.60 * × 10^4^
PRS	(2–5 months)	Male	2.18 × 10^6^ ± 11.9 × 10^4^	2.11 × 10^6^ ± 15.4 × 10^4^
		Female	2.04 × 10^6^ ± 9.11 × 10^4^	2.39 × 10^6^ ± 9.11 × 10^4^
	(12–16 months)	Male	2.29 × 10^6^ ± 6.68 × 10^4^	2.39 × 10^6^ ± 20.0 × 10^4^
		Female	2.11 × 10^6^ ± 22.2 × 10^4^	3.07 × 10^6^ ± 13.1 × 10^4^
CA2	(2–5 months)	Male	0.54 × 10^6^ ± 7.99 × 10^4^	0.79 × 10^6^ ± 9.11 × 10^4^
		Female	0.64 × 10^6^ ± 9.11 × 10^4^	0.61 × 10^6^ ± 4.37 × 10^4^
	(12–16 months)	Male	0.75 × 10^6^ ± 6.68 × 10^4^	0.71 × 10^6^ ± 12.6 × 10^4^
		Female	0.68 × 10^6^ ± 6.68 × 10^4^	0.86 × 10^6^ ± 10.4 × 10^4^

**Table 2 ijms-24-14917-t002:** Table gives actual values of all neuroanatomy data from immunofluorescence analysis of the CA1 region. All values are stated as mean ± SEM. (* *p* < 0.05, *n* = 5 animals per cohort) between wild-type control and *APP ^NLF/NLF^* values.

Brain Region	Age	Parameter	Wild-Type		*APP^NL-F/NL-F^*	
			Male	Female	Male	Female
			Mean ± SEM	Mean ± SEM	Mean ± SEM	Mean ± SEM
CA1	12–16 months	Aβ plaque size	25.46 ± 6.23	41.22 ± 4.39	361.38 ± 76.80 *	359.31 ± 53.05 *
		CD68+ density	15.66 ± 1.64	14.22 ± 3.69	35.33 ± 1.92 *	38.71 ± 1.72 *
		PNN density	3.48 ± 0.88	2.71 ± 0.20	13.34 ± 1.78 *	7.66 ± 0.99 *

**Table 3 ijms-24-14917-t003:** List of antibodies and dilutions used in this study.

Antibody	Target	Manufacturer	Dilution	Host
Amylo-Glo^®^ RTD^TM^	Amyloid Plaques	Avantor	1:100	N/A
CD68 Monoclonal	Mouse microsialin	Bio-Rad	1:3000	Rat
Wisteria Floribunda Lectin (WFA, WFL), Biotinylated	*N*-acetyl galactosamine linked α or β terminal end of carbohydrates to the 3 or 6 positions of galactose	Vector Laboratories	1:400	N/A
Goat anti-rat antibody Alexa 488	_______________	Invitrogen	1:500	Goat

## Data Availability

The data presented in this study are available on request from the corresponding author. The data are not publicly available due to privacy.

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
