# Peer review of "Age-Dependent Sex Differences in Perineuronal Nets in an APP Mouse Model of Alzheimer’s Disease Are Brain Region-Specific"

_ijms, 2023, doi:10.3390/ijms241914917_

Round 1

Reviewer 1 Report

The authors try to demonstrate the disrupted PNNs and microglia interaction in specific brain region causing sex differences in AD pathogenesis by using an AD mouse model. But the evidence is weak.

1. The authors should also show the images of overall PNN expression pattern in whole brain sections.

2. The graphs of Figure 2C, 2F, 2I show statistically significant differences but the images in figure 2A, 2D, 2G did not show obvious differences. The authors should use more representative images and provide detailed quantification protocol in methods.

This manuscript presented an interesting study and well-written. But several points should be addressed before publishing.

1. Figure resolution should be improved. The text and images are blurred.

2. Scale bar is missing in Figure 1 A and B.

3. The text for scale bar is too small in Figure 2A, 2D, 2G. It can be label outside of images.

Author Response

“The authors try to demonstrate the disrupted PNNs and microglia interaction in specific brain region causing sex differences in AD pathogenesis by using an AD mouse model. But the evidence is weak.

  1. The authors should also show the images of overall PNN expression pattern in whole brain sections.”

Response: We thank the reviewer for their great suggestions and have now included a supplementary figure S1 that illustrates the staining of the PNNs in the whole brain sections for each region. This has also been included in the text (Pages 9 and 13) to deatil the location of the brain region with guidance from the mouse brain atlas (rain Map - brain-map.org [Internet]. [cited 2023 Sep 19]. Available from: https://portal.brain-map.org/ ).    

“2. The graphs of Figure 2C, 2F, 2I show statistically significant differences but the images in figure 2A, 2D, 2G did not show obvious differences. The authors should use more representative images and provide detailed quantification protocol in methods.”

Response: We thank the reviewer for their valuable feedback. The resolution of our images has now been improved as per reviewer’s suggestion. Also, a detailed description of the quantification method is mentioned in the methods and materials section on pages 9-10.

Furthermore, we have replaced various images in Figure to to show better representation of the quantitative data illustrates in 2C, 2F and 2I.

This manuscript presented an interesting study and well-written. But several points should be addressed before publishing.

  1. Figure resolution should be improved. The text and images are blurred.

Response: We apologise for the change in the quality and resolution of the images when uploading to the manuscript website. We are concerned about our image quality, and therefore to ensure optimum quality, we have converted each image at high resolution of 300DPI and converted this to a tiff.

We will make these images available to the editor as a ZIP file once the manuscript has been accepted. We have uploaded the newer versions, which still may be distorted due to the various conversion stages.

  1. Scale bar is missing in Figure 1 A and B.

Response: We apologise for omitting this information, and we greatly appreciate the reviewer for bringing this to our attention. All scale bars have been adjusted for the figures, and they have been changed to a different colour (RED) to make them more distinguishable. The length of the scale bars is also included in each figure legend accordingly.  

  1. The text for scale bar is too small in Figure 2A, 2D, 2G. It can be label outside of images.

Response: The text on the scale bars has been removed, and as mentioned in the previous response, scale bars have now been changed to a different colour to improve their visuality, and details of scale bar lengths is now included in the figure legends.

Reviewer 2 Report

This manuscript by Rayane Rahmani and co-workers dissects the question how does aging and sex affect the perineural net in normal and genetically modified mice that develop an Alzheimer disease-like state.

The morphological results are interesting; the study design is well planned. This reviewer is on the point of view that the data are worth to publish in IJMS, however Authors may treat the following points in order to increase the scientific value of the manuscript.

1. Statistics. Authors claim that they performed two-way ANOVA-s. The two factors were age and sex.

a) It would important to see the main effects of these including F and p values, with the degrees of freedom and denominator error.

b) Also here, in order to see whether the basis of post hoc comparisons is given, the interactions should also be described.

c) ANOVA tests require normal distribution of data and the homogeneity of variance. However, these kind of morphological datasets often show lognormal distribution. How was the normality and homogeneity tested? What did Author do if these pre-requisites of the ANOVA were violated?

2. Image quality. The reviewer has the impression that Authors did not insert their high resolution images into the manuscript, or the PDF conversion reduced the resolution too much. As this is a nice morphological study, Authors are advised to find a way to increase the resolution of their figures. Higher magnification inserts would also be helpful.

3. The signal density / intensity of the PNN labeling is ofter referred to as "expression" throughout. This word is not the right choice in this context as this is not about mRNA expression in the cells, but the abundance / occurrence / intensity of a signal represents the presence/amount of in the perineural net constituents. Authors may rephrase this throughout.

4. Related to comment No 3, Authors in the abstract write about "microglia expression" that is also not properly formulated. Microglia cells may express mRNAs, may contain proteins, may show stronger or weaker CD68 immunoreactivity, but this should be clearly stated. 

5. Authors in the figure legends claim that they applied 10x or 20x magnifications. This is however the magnification of the objective lens that they used to take the images. The final magnification is calculated by multiplying the objective lens magnification with the magnification of the ocular lens, or in this case, that of the camera lens.

6. Authors forgot to define the length of scale bars in the legends of the histological images.

7. Authors, in the figure legends do not describe what p value does the labeling “****” (four asterisks) refer to.

8. Authors describe and define the short terms for the hippocampal layers, but they do not appear in the figures, or they are that small that one cannot read is. This seems to be superfluous, or the figure resolution and size should be adjusted that they get readable.  

9. The use of blue and red symbols is not consistent, and it is therefore very confusing. In most parts of the World, the red/pink color is associated with females/women, and the blue refers to males/men. Therefore, this reviewer would suggest to change the grey colors of the graph bars accordingly. More importantly, the color of data point triangles and circles may be re-colored accordingly (e.g. red circle - female; blue triangle -  male).

10. The first paragraph of the Results section (ln 103-116) contains redundant parts in comparison with the introduction. Because in this journal the Methods part does not separate the Results from the Introduction part, this paragraph seems to be superfluous and can be deleted, or radically shortened.

11. Authors are asked to double-check the abbreviations systematically. They are sometimes not defined at all (e.g. CSPGs) are not defined at the first use, but just later, and in other cases some short terms are defined, but they do not come up again, again others are defined twice (e.g. ARM).

12. Pg. 3. ln 125-126. n=5 comes twice.

13. ln. 444. The nanometer should be corrected to nm instead of lambda.  

Authors write sometimes too long sentences. Some sentences may be rephrased (e.g. pg. ln 68 "...shows disappeared Ab", ln 301-302). The text contains several double spaces. The punctuation requires also more attention at the end of sentences. Region-specific (hyphen missing).

Author Response

This manuscript by Rayane Rahmani and co-workers dissects the question how does aging and sex affect the perineural net in normal and genetically modified mice that develop an Alzheimer disease-like state.

The morphological results are interesting; the study design is well planned. This reviewer is on the point of view that the data are worth to publish in IJMS, however Authors may treat the following points in order to increase the scientific value of the manuscript.

  1. Statistics. Authors claim that they performed two-way ANOVA-s. The two factors were age and sex.
  2. a) It would important to see the main effects of these including F and p values, with the degrees of freedom and denominator error.
  3. b) Also here, in order to see whether the basis of post hoc comparisons is given, the interactions should also be described.
  4. c) ANOVA tests require normal distribution of data and the homogeneity of variance. However, these kinds of morphological datasets often show lognormal distribution. How was the normality and homogeneity tested? What did Author do if these pre-requisites of the ANOVA were violated?

Response to a-c: We greatly appreciate the reviewer’s comments and we have now clarified the stats performed further, included the F and p values of our two-way ANOVA in the methods, pages 12:

“Before performing any statistical test, the normality of the raw data was verified using the test Shapiro-Wilk and a ROUT test to identify potential outliers. In almost all of the cases no outliers were identified. Where an outlier existed, it was removed from the data pool via the software. When comparing between three or more data sets from two genotypes, a one-way analysis of variance (ANOVA) was performed. When a second factor was taken into account, for example age, a two-way ANOVA was used. After any ANOVA, a post-hoc test for multiple comparisons was applied. All P-values below 0.05 were considered significant and asterisks added to the presentation of the data as follows: * p<0.05 ** p<0.01 *** p<0.001 **** P<0.0001. F-statistical values were calculated by the Prism software following ANOVA analysis, these values were also used to assess differences between groups, higher F values, generally F>8 were considered significant.”

  1. Image quality. The reviewer has the impression that Authors did not insert their high resolution images into the manuscript, or the PDF conversion reduced the resolution too much. As this is a nice morphological study, Authors are advised to find a way to increase the resolution of their figures. Higher magnification inserts would also be helpful.

Response: We apologise for the change in the quality and resolution of the images when uploading to the manuscript website. We are concerned about our image quality, and therefore to ensure optimum quality, we have converted each image at high resolution of 300DPI and converted this to a tiff.

We will make these images available to the editor as a ZIP file once the manuscript has been accepted. We have uploaded the newer versions, which still may be distorted due to the various conversion stages.

  1. The signal density / intensity of the PNN labeling is ofter referred to as "expression" throughout. This word is not the right choice in this context as this is not about mRNA expression in the cells, but the abundance / occurrence / intensity of a signal represents the presence/amount of in the perineural net constituents. Authors may rephrase this throughout.

We have now revised the manuscript thoroughly and change the word expression to PNN density / PNN abundance for more appropriation of the term.

  1. Related to comment No 3, Authors in the abstract write about "microglia expression" that is also not properly formulated. Microglia cells may express mRNAs, may contain proteins, may show stronger or weaker CD68 immunoreactivity, but this should be clearly stated. 

Response: We have clarified this in the abstract and changed this to “Aβ plaques near clusters of CD68, indicative of activated microglia”.

  1. Authors in the figure legends claim that they applied 10x or 20x magnifications. This is however the magnification of the objective lens that they used to take the images. The final magnification is calculated by multiplying the objective lens magnification with the magnification of the ocular lens, or in this case, that of the camera lens.

Response: We apologise for this oversight which has now been amended accordingly to the following:

Page 21, figure 1 legend: “brightfield images taken at x100 magnification”

Page 22, figure 2 legend: “images taken at x200 magnification”

Page 23, figure 3 legend: “at X200 magnification”

  1. Authors forgot to define the length of scale bars in the legends of the histological images.

Response: We sincerely apologize for omitting this information which has now been added to the figure legends in the manuscript:

On page 21, figure 1 legend: Scale bar 20 µm.

Page 22, figure 2 legend: Scale bar 10 µm.

Page 23, figure 3 legend: Scale bar 50 µm

  1. Authors, in the figure legends do not describe what p value does the labeling “****” (four asterisks) refer to.

Response: We greatly thank the reviewer for pointing this out and we apologize for omitting this information. This has now been added to the figure legends accordingly. On pages 22, line 511; page 23, line 523; and page 24, line 538. To “****P<0.0001.”

  1. Authors describe and define the short terms for the hippocampal layers, but they do not appear in the figures, or they are that small that one cannot read is. This seems to be superfluous, or the figure resolution and size should be adjusted that they get readable.  

Response: We apologize for the confusion this has caused and thank the reviewer for their valuable comment. The short terms are indeed too small which made it difficult for the reader to see. The labels on the figure have now been removed to provide better clarity of the figure, and the hippocampus layers that were analyzed have now been highlighted in the methods and materials section on page 10 as follows:

“CA1 (stratum oriens, pyramidal layer, and stratum radiatum), LEC (layer 2 and 3), PRS (layers 1 and 2),”

  1. The use of blue and red symbols is not consistent, and it is therefore very confusing. In most parts of the World, the red/pink color is associated with females/women, and the blue refers to males/men. Therefore, this reviewer would suggest to change the grey colors of the graph bars accordingly. More importantly, the color of data point triangles and circles may be re-colored accordingly (e.g. red circle - female; blue triangle -  male).

Response: We apologise for the confusion; there many parameters and we have refined our “key” in each figure. Trianges denote males, whereas circles denotes data points for females. The different shades of grey refer to the genotype.

  1. The first paragraph of the Results section (ln 103-116) contains redundant parts in comparison with the introduction. Because in this journal the Methods part does not separate the Results from the Introduction part, this paragraph seems to be superfluous and can be deleted, or radically shortened.

Response: We greatly appreciate the reviewer’s invaluable feedback. This paragraph has been shortened in page 13, lines 246-249 to the following:

“The current study aimed to investigate whether there was an alteration of sub-brain region intensity of WFA, representing PNNs in early (2-5 months) and late-stage (12-16 months) AD using an APP knock-in mouse model compared to wild-type litter mates, and whether there was a sex-related difference in the density of PNNs.”

  1. Authors are asked to double-check the abbreviations systematically. They are sometimes not defined at all (e.g. CSPGs) are not defined at the first use, but just later, and in other cases some short terms are defined, but they do not come up again, again others are defined twice (e.g. ARM).

Response: We have now updated the manuscript, making sure all abbreviations are defined appropriately. For example, in page 5 Line 80, CSPGs are defined during the first use of the word e.g., “creating a netlike structure composed of a hyaluronan (HA) backbone linked to chondroitin sulphate proteoglycans (CSPGs)”

  1. Pg. 3. ln 125-126. n=5 comes twice.

Response: The doubling of “n=5” has now been removed in page 13 lines 260 and 264

  1. ln. 444. The nanometer should be corrected to nm instead of lambda.  

Response: Our sincere apologies for this mistake, and have changed the lambda symbol to nm in page 10 lines 205-206 accordingly as follows:

“DAPI (405 nm), Alexa 488 (488 nm), Alexa 555 (555/565 nm).”

Comments on the Quality of English Language

  1. Authors write sometimes too long sentences. Some sentences may be rephrased (e.g. pg. ln 68 "...shows disappeared Ab", ln 301-302). The text contains several double spaces. The punctuation requires also more attention at the end of sentences. Region-specific (hyphen missing).

Response: We hope the reviwer finds the revised version of the manuscript acceptable and we thank them for their valuable feedback which has improved the flow of the manuscript.

During revision, long sentences have now been revised throughout, for exmaple:

Page 6 line 94-95 as follows: despite the neighbouring Presubiculum (PRS) region which was found to be an unscathed region in clinical AD.

The punctuation and grammar have also been reviewed throughout the manuscript and are now hopefully up to a good standard.

Reviewer 3 Report

Rahmani et al. investigated whether PNNs are altered during early and late-stage AD and whether differences in the abundance of PNNs in AD-vulnerable regions were associated with differences in sex. This is an interesting paper, indicating that the expression of PNNs is brain region-specific, with the highest PNN abundance in the LEC and PRS regions, and significant sex differences in the LEC, PRS and the CA1 region. Furthermore, the authors suggested that sex differences contribute to a disrupted interaction between PNNs and microglia in specific brain regions associated with AD pathogenesis. There are, however, several issues to be addressed to further improve the manuscript.

1.     In the hippocampal region, WFA positive PNNs are particularly densely distributed in CA2 region compared to CA1 and CA3 regions. The hippocampal CA2 region plays a critical role in social recognition memory and has unique cellular and molecular properties that distinguish it from areas CA1and CA3, which means CA2 region could be altered during AD. Thus, in addition to CA1 region, it is strongly recommended to investigate whether PNNs in the CA2 region were altered during early and late-stage AD and associated with difference in sex.

2.     In Fig.3A, sex-related changes in microglial morphology cannot be detected. More clear and higher magnification images with higher resolution should be required.

3.     As the authors themselves mentioned at lines 167-174, z-stack images from immunofluorescence labelling provided a better depth of tissue area analyzed for providing more precise data. Thus, the data obtained from immunoperoxidase labelling shown in Fig. 2C should be confusing.

4.     In CA1 region, it is required to investigate whether PNNs positive neurons express estrogen receptors and compare the co-expression rates between male and female.

5.     What do the white arrows in Figure 3 point to?

Author Response

Rahmani et al. investigated whether PNNs are altered during early and late-stage AD and whether differences in the abundance of PNNs in AD-vulnerable regions were associated with differences in sex. This is an interesting paper, indicating that the expression of PNNs is brain region-specific, with the highest PNN abundance in the LEC and PRS regions, and significant sex differences in the LEC, PRS and the CA1 region. Furthermore, the authors suggested that sex differences contribute to a disrupted interaction between PNNs and microglia in specific brain regions associated with AD pathogenesis. There are, however, several issues to be addressed to further improve the manuscript.

  1. In the hippocampal region, WFA positive PNNs are particularly densely distributed in CA2 region compared to CA1 and CA3 regions. The hippocampal CA2 region plays a critical role in social recognition memory and has unique cellular and molecular properties that distinguish it from areas CA1and CA3, which means CA2 region could be altered during AD. Thus, in addition to CA1 region, it is strongly recommended to investigate whether PNNs in the CA2 region were altered during early and late-stage AD and associated with difference in sex.

Response: We greatly appreciate the reviewer’s feedback and have now included the following:

  1. the results of PNN density in the CA2 region, noted on page 13
  2. the data for PNN density for CA2 on Table 2
  3. and supplementary figure S2 to show the labelling of PNNs

Page 13: “The results below detail the analysis performed in the LEC, PRS and the CA1 region of the hippocampus which revealed interesting findings. However, we also imaged the density of PNNs in the hippocampal CA2 region of the hippocampus, that is thought to play a critical role in social recognition memory and has unique cellular and molecular properties that distinguish it from areas (Rey et al., 2022). The density of the PNN labelling in CA2 was consistent with the different age and genotype suggesting no significant alteration during the pathogenesis of AD and thus has been included as supplementary material (S2). The magnitude of the PNN density is included in Table 2 where a direct comparison with the other brain regions can be observed.”

  1. In Fig.3A, sex-related changes in microglial morphology cannot be detected. More clear and higher magnification images with higher resolution should be required.

Response: We appreciate the valuable feedback here for the IF analysis in Figure 3A, which showed no significant sex related differences between 12-16 months males and females and will consider incorporating further work using high resolution imaging into our future work. Although we have not directly investigated microglial “morphology” in detail, we have included this aspect in out discussion (page 20, line 4487- 449).

  1. As the authors themselves mentioned at lines 167-174, z-stack images from immunofluorescence labelling provided a better depth of tissue area analyzed for providing more precise data. Thus, the data obtained from immunoperoxidase labelling shown in Fig. 2C should be confusing.

Response: We appreciate this concern of the reviewer, and have incorporated two different anatomical techniques to validate our data. We would appreciate any further feedback or instructions on how to resolve this issue.

Where possible, we have clarified any confusion related to immuneperoxidase V immunofluorescence staining.  

  1. In CA1 region, it is required to investigate whether PNNs positive neurons express estrogen receptors and compare the co-expression rates between male and female.

We thank the reviewer for their helpful comments and recognize that the connections between PNNs and Oestoogens is an important one and our study limitation. To study the connection between PNNs and oestrogens is beyond the scope of this study and will take a further 18 months to perform, which would be a great Master project for a student no doubt.

However, we have included further discussion on this theme, clearly indicating our limitations and further investigations that could be carried out.  We have also cited the work of others in this arena. The closest study is by Blurton-Jones et al., 2002, that show that these receptors are highly co-expressed on PV cells, thus connection with the PNNs.

Changes to discussion, pages 19:

“is perhaps a connection between PNNs and oestrogens which is understudied. This corroborates previous study that shows PNN associated neurons in the mPOA (medial Preoptic area) express oestrogen receptor α which is suggested to influence the signalling mechanism affecting the assembly of the PNNs during the reproductive cycle in rodents (Uriarte et al., 2020).

 Evidence also suggests a high density of oestrogen receptors in the hippocampus (Spencer-Segal et al., 2012), and interestingly they are colocalised extensively with PV cells in various brain region including the hippocampus (Blurton-Jones and Tuszynski, 2002). This suggests a potential connection of oestrogen with PNNs that encapsulate PV cells, and a mechanism whereby oestrogen can regulate neuronal excitability via inhibitory interneurons, which aligns with previous studies that hypothesis oestrogen receptors to  influences hippocampal synaptic plasticity and neural signalling, that  may affect cognition and memory (McEwen et al., 2012).

Furthermore, Oestrogen  are thought to have several  roles in AD (Li and Singh, 2014), including neuroprotection and influence the regulation of the amyloid beta pathology by reducing the levels of Ab and preventing its accumulation (Anastasio, 2013), which are functions comparable to that of PNNs (Reichelt et al., 2019). Therefore, it may be plausible that males require a higher abundance of PNNs in brain regions such as the CA1 to obtain neuroprotection against factors such as oxidative stress which they are more susceptible to (Griffiths et al., 2019), as they do not have the potential extra protection that can be provided by the oestrogen hormones. The lack of experimental connection between PNNs and Oestrogens in the current study is a limitation of the current study, and would need to be investigated further and could potentially prove as a novel approach to understanding the sex related differences in AD prevalence.”

  1. What do the white arrows in Figure 3 point to?

Response: We sincerely apologise for the lack of clarity regarding figure 3, and for omitting this information. The white arrows in the figure point to the amyloid beta plaques in the CA1 region of the APP mouse model. This has now been included in the figure legend for figure 3 in page 24 line 477-478

“The white arrows in figure A point to Amyloid beta plaque in APP mouse model”

Round 2

Reviewer 2 Report

Authors, when revising their paper have performed some modification in the manuscript, but they ignored or did not perform most of the major concerns of this reviewer.

1. Statistics.

Authors do not describe clearly where did they use the one-way ANOVA, and where was the two-way ANOVA done.

Authors ignored the suggestion to add the F values with the degrees of freedom and denominator error.

Authors claim in the revised version that the "higher F values, generally F>8 were considered significant". This scientifically hardly understandable statement does not suggest that the Authors performed their statistics in a proper professional way. 

This reviewer would insist to add the proper F and p values for each calculation, including the F and p values for main effects and in case of two way ANOVAS also the F and the p values of interactions because without theseit is impossible to judge whether the reliability of post hoc tests is supported by the ANOVA values.

2. Image quality. Authors claim that they increased the resolution of their images but they also state that this may be invisible as the pdf conversion may reduce the quality again. However, in a morphological work, the morphology must be well visible and convincing. This reviewer can only assess the figures that are in the manuscript, and as they appear now they are still too blurry and not sharp enough. Some examples: Fig1. 12-16 months LEC, WT female is fuzzy, but all other LEC images in this figure look more or less blurry. Even in Fig 4, that is a table with text, one cannot read the text at all. Authors may probably ask the technical experts of image quality at the journal to find a way to fix these general problems on figure quality, because in many cases the reader simply cannot see in the pictures that Authors want to show. 

3. Authors also ignored to remove the "PNN expression" or "expression of PNN". The first comes up 5 times just in the abstract, the latter one 7 times in figure legends and 15 times in the text.

4. Authors did not remove the wording "expression of microglia" either (abstract, line 14).

6. Scale bars still missing in fig S1. In Fig S2 the newly added red scale bars were placed above the old black ones, but the black one remained visible as the red one does not cover them (e.g. PRS 2-5 months old female APP mouse image, an in other images/panels).

9. Authors still don't use the symbols for females and males systematically. In Fig 1, blue sometimes refers to females, sometimes to males. This, even afer revision, remained very confusing. 

11. Authors use the short term PV twice, but as it is never defined in the manuscript, one does not really know if it perhaps to parvalbumin.

Author Response

Ref 2 Round 2: Changes to the manuscript are highlighted.

Authors, when revising their paper have performed some modification in the manuscript, but they ignored or did not perform most of the major concerns of this reviewer.

  1. Statistics.

Authors do not describe clearly where did they use the one-way ANOVA, and where was the two-way ANOVA done.

Authors ignored the suggestion to add the F values with the degrees of freedom and denominator error.

Authors claim in the revised version that the "higher F values, generally F>8 were considered significant". This scientifically hardly understandable statement does not suggest that the Authors performed their statistics in a proper professional way. 

This reviewer would insist to add the proper F and p values for each calculation, including the F and p values for main effects and in case of two way ANOVAS also the F and the p values of interactions because without theseit is impossible to judge whether the reliability of post hoc tests is supported by the ANOVA values.

Response: We have now included the F and P values as suggested by the review and have changed this firstly in the methods, page 12, line 250:

“To assess the significance of the differences between groups, F-values were also taken into consideration. There are displayed as (F values(dfn,dfd) ; p value), where dfn =  for degrees of freedom- numerator, and dfd stands for degrees of freedom denominator.  As for the interpretation, larger F values were considered significant, whereas smaller F values were considered non-significant. “

Secondly, we included these values throughout the results, e.g. page, 13, line277

“2-5 months wild-type (F2,24 = 131.2; p<0.0001), and APP mice (F2,24 = 120.8; p<0.0001), as well as 12-16 months old wild-type (F2,24 = 83.90; p<0.0001), and APPNL-F/NL-F mice (F2,24 = 79.94; p<0.0001) (Figure 1).  “

  1. Image quality. Authors claim that they increased the resolution of their images but they also state that this may be invisible as the pdf conversion may reduce the quality again. However, in a morphological work, the morphology must be well visible and convincing. This reviewer can only assess the figures that are in the manuscript, and as they appear now they are still too blurry and not sharp enough. Some examples: Fig1. 12-16 months LEC, WT female is fuzzy, but all other LEC images in this figure look more or less blurry. Even in Fig 4, that is a table with text, one cannot read the text at all. Authors may probably ask the technical experts of image quality at the journal to find a way to fix these general problems on figure quality, because in many cases the reader simply cannot see in the pictures that Authors want to show. 

Response: We very much appreciate this comment from the review and are sincerely apologetic, however, some issues surrounding technical difficulties is beyond our control. You will see in the image below that we have indeed converted our graphics files to high resolution 300 dpi. The images do not appear pixelated when we zoom in. From experience the figure resolution issues are resolved during fine proof editing by the journal once we have submitted our original files. We assure the reviewer that we will work with the publication team to ensure high quality images are published which is our goal.

  1. Authors also ignored to remove the "PNN expression" or "expression of PNN". The first comes up 5 times just in the abstract, the latter one 7 times in figure legends and 15 times in the text.

Response: We sincerely apologise to the reviewer for omitting this. We have reviewed the text, abstract and figure legends, and made sure that the word expression is removed throughout.

  1. Authors did not remove the wording "expression of microglia" either (abstract, line 14).

Response: Again, we apologise sincerely for omitting this. “The expression of microglia” has now been changed to “labelling of microglia” on page 2, line 28. We hope that this is a more appropriate wording.

  1. Scale bars still missing in fig S1. In Fig S2 the newly added red scale bars were placed above the old black ones, but the black one remained visible as the red one does not cover them (e.g. PRS 2-5 months old female APP mouse image, an in other images/panels).

Response: We apologize for this and have now added scale bars in S1 and included details in the legend. S2 does not show PRS, but Figure 2 has been checked and corrected.

  1. Authors still don't use the symbols for females and males systematically. In Fig 1, blue sometimes refers to females, sometimes to males. This, even afer revision, remained very confusing. 

Response: We have changed figure 1 symbols this round, now throughout the manuscript (despite the genotype), blue triangles = males, red circles = females. Thank you for helping us improve the manuscript.  

  1. Authors use the short-term PV twice, but as it is never defined in the manuscript, one does not really know if it perhaps to parvalbumin.

Response: We apologise for this.  The definition of PV indeed refers to parvalbumin, which has now been defined in Page 19, line 422 as follows: “with parvalbumin (PV) cells”.

Reviewer 3 Report

The authors put an effort in revising their manuscript and addressing issues raised previously. The paper was improved. However, there is still one concern. Although the authors mentioned that the data for PNN density for CA2 on Table 2, this data cannot be found in Table 2.

Author Response

Response:

We thank the reviewer for their kind words regarding the improvement of the paper. Regarding the CA2 data, it has already been inserted in table 2 during the first revision, on page 29, line 538. The data has now been highlighted.

CA2

(2-5 months)​

Male​

0.54 x 106 ±  7.99 x 104

0.79 x 106 ±  9.11 x 104

Female​

0.64 x 106 ±  9.11 x 104

0.61 x 106 ±  4.37 x 104

(12-16 months)​

Male​

0.75 x 106 ±  6.68 x 104

0.71 x 106 ±  12.6 x 104

Female​

0.68 x 106 ±  6.68 x 104

0.86 x 106 ±  10.4 x 104

Round 3

Reviewer 2 Report

Authors have addressed all the critial remarks of this reviewer adequately. When reading the latest version of the mansucript, no new critial issues came up. Authors are advised to check the quality of figures in the proof to make sure that the resolution of figures in the published final version will be the best/optimal.